# Pharmacokinetic/Pharmacodynamic (PK/PD) Simulation for Dosage Optimization of Colistin Against Carbapenem-Resistant *Klebsiella pneumoniae* and Carbapenem-Resistant *Escherichia coli*

**DOI:** 10.3390/antibiotics8030125

**Published:** 2019-08-22

**Authors:** Kamonchanok Jitaree, Korbtham Sathirakul, Jantana Houngsaitong, Orarik Asuphon, Weerayuth Saelim, Visanu Thamlikitkul, Preecha Montakantikul

**Affiliations:** 1Department of Pharmacy, Faculty of Pharmacy, Mahidol University, Bangkok 10400, Thailand; 2Department of Pharmacy Practice, Faculty of Pharmaceutical Sciences, Naresuan University, Phitsanulok 65000, Thailand; 3Department of Pharmacy, Faculty of pharmacy, Silpakorn University, Nakhon Pathom 73000, Thailand; 4Division of Infectious Disease and Tropical Medicine, Faculty of Medicine, Siriraj Hospital, Mahidol University, Bangkok 10700, Thailand; 5Department of Pharmacy, Faculty of Pharmacy, Mahidol University, Bangkok 10400, Thailand

**Keywords:** colistin, Monte Carlo simulation, pharmacokinetics, pharmacodynamics, carbapenem-resistant *Klebsiella**pneumoniae*, carbapenem-resistant *Escherichia coli*

## Abstract

The purpose was to explore the optimal dosage regimen of colistin using Monte Carlo simulations, for the treatment of carbapenem-resistant *Klebsiella pneumoniae* and carbapenem-resistant *Escherichia coli* based on PK/PD targets in critically ill patients. A total of 116 carbapenem-resistant *K. pneumoniae* and *E. coli* were obtained from various clinical specimens at Siriraj Hospital in Bangkok, Thailand. Minimum inhibitory concentrations (MICs) of colistin were determined by broth microdilution method. Monte Carlo simulation was used to calculate the cumulative fraction of response (CFR) for European Medicine Agency (EMA), US-Food and Drug Administration (FDA), Nation et al., Siriraj Hospital and our study regimens. The targeted CFR was 90%. For colistin-susceptible *K. pneumoniae*, all of the dosage regimens achieved ≥90% CFR in patients with creatinine clearance <80 mL/min except the FDA-approved regimens for patients with creatinine clearance 51–79 and 11–29 mL/min, respectively. While, patients with creatinine clearance ≥80 mL/min, CFR ≥90% was observed in Siriraj Hospital and our study regimen. For colistin-susceptible *E. coli*, all of the dosage regimens achieved ≥90% CFR regardless of renal function. In contrast, the currently approved regimens achieved CFR target in only 10-50% for colistin-resistant isolates subgroup. These results suggest that currently approved regimens still recommended for colistin-susceptible CRE. For colistin-resistant CRE, alternative approaches such as high dose or combination therapy should be considered.

## 1. Introduction

Worldwide emergence of carbapenem-resistant gram-negative bacteria, and particularly carbapenem-resistant Enterobacteriaceae (CRE) have become a major concern [1]. Specifically, carbapenem-resistant *Klebsiella pneumoniae* and carbapenem-resistant *Escherichia coli* which most commonly cause hospital acquired infections like intra-abdominal infections, urinary tract infections, nosocomial pneumonia, and bloodstream infections with high mortality rate of 44% to 70% [2,3]. In Thailand, the prevalences of carbapenem-resistant *K. pneumoniae* as well as carbapenem-resistant *E. coli* were approximately 2–10% in 2017 and these had rapidly increased from 0.3–0.8% in 2010 [4,5,6]. Concomitantly, there has been a shortage of newer antibiotics to fight multidrug-resistant (MDR) bacteria development. This has led to re-emergence of previously used old antibiotics like colistin.

Colistin has been developed as a mainstay of treatment for patients infected by carbapenem-resistant Enterobacteriaceae [7,8]. Although, optimal dosage regimen for colistin has been unclear because accurate pharmacodynamic and pharmacokinetic information are lacking. However, recent studies have improved our understanding about the pharmacokinetic/pharmacodynamic (PK/PD) which has guided us to selecting suitable colistin dosage regimen to treat carbapenem-resistant Enterobacteriaceae infections [7,8,9]. Suitable dosing strategies has lead to maximize the clinical efficacy, and to reduce the developing bacterial resistance rate, morbidity and mortality.

This study aimed to explore the optimal dosage regimen of colistin using Monte Carlo simulations, for the treatment of carbapenem-resistant *K. pneumoniae* and carbapenem-resistant *E. coli* based on PK/PD targets in critically ill patients.

## 2. Materials and Methods

### 2.1. Microbiology

Carbapenem-resistant *K. pneumoniae* and carbapenem-resistant *E. coli* were obtained from various clinical specimens (sputum, urine, skin and soft tissue, blood, and pleural fluid) which were submitted for culture to the microbiological laboratory at the Faculty of Medicine at Siriraj Hospital in Bangkok, Thailand. The isolates were collected between June 2011 and September 2016. The Carbapenem-resistant Enterobacteriaceae (CRE) phenotype was identified for isolates expressing non-susceptible (intermediate or resistant) to any carbapenem antibiotics or documented to produce carbapenemase. Minimum inhibitory concentrations (MICs) of colistin were determined by the broth microdilution method in cation-adjusted Mueller-Hinton II broth (CA-MHBII) according to Clinical and Laboratory Standards Institute (CLSI) guidelines [10]. Colistin was tested over a range of dilutions from 0.25 mcg/mL to 128 mcg/mL. *Escherichia coli* ATCC 25922 was used as a control. Isolate preparation was performed following the Clinical and Laboratory Standards Institute protocol (CLSI) [10]. The MIC value was defined as the lowest drug concentration for which the well was optically clear. The breakpoints of the European Committee on Antimicrobial Susceptibility Testing (EUCAST) (MIC ≤ 2 mcg/mL) were applied to interpret those for colistin [11].

### 2.2. Pharmacokinetic Model

The population pharmacokinetic model for colistimethate sodium (CMS) and colistin were two-compartment and one-compar tment models, respectively. Pharmacokinetic data from Nation et al. [12] was retrieved from previously published studies concerning critically ill patients. Table 1 A set of parameters was randomly generated for each mean and standard deviation of the parameters. Equations (1)–(3) below represent the differential equations for the disposition of CMS and colistin.
(1)$$\frac{\mathrm{dCMSc}}{\mathrm{dt}}=R1-\mathrm{CLD}1X\left(\frac{\mathrm{CMSc}}{V1}-\frac{\mathrm{CMSp}}{V2}\right)-\left(\mathrm{CLTcms}\;X\frac{\mathrm{CMSc}}{V1}\right)$$
(2)$$\frac{\mathrm{dCMSp}}{\mathrm{dt}}=\mathrm{CLD}1\;X\left(\frac{\mathrm{CMSc}}{V1}-\frac{\mathrm{CMSp}}{V2}\right)\;$$
(3)$$\frac{\mathrm{dColistin}}{\mathrm{dt}}=\left(\mathrm{CLNRcms}\;X\frac{\mathrm{CMSc}}{V1}\right)-\left(\mathrm{CLTc}\;X\frac{\mathrm{Colistin}}{V3}\right)\;$$

CMSc = mass of CMS in the central compartment, CMSp = mass of CMS in the peripheral compartment, Colistin = mass of colistin in the single compartment, V1 = central volumes for CMS, V2 = peripheral volumes for CMS, V3 = volume of distribution for colistin, R (1) = infusion rate of CMS, CLD1 = distributional clearance between the central and peripheral compartments for CMS. CLTcms = total intrinsic clearance for CMS, CLTc = total intrinsic clearance for colistin and CLNRcms = nonrenal clearance for CMS.

### 2.3. Pharmacodynamic Model

The pharmacodynamic (PD) surrogate indices for colistin are characterized by area under the unbound colistin plasma concentration-time curve to the minimum inhibitory concentration (*f*AUC/MIC) [13,14]. PK/PD targets were defined as *f*AUC/MIC ≥25 [13]. The trapezoidal rule was used to calculate AUC, then *f*AUC was divided by the MIC to obtain the value for the desired PK/PD index. A colistin unbound fraction of 50% was used [15]. The regimen administrated using IV infusions (infused over 30 min) were simulated over a 24-h period at different levels of renal function. The ranges of creatinine clearance (CrCl) used were ≥80, 51 to 79, 30 to 50, 11 to 29 and ≤10 mL/min. The other dose was shown in Appendix A.

The regimens were chosen by using European Medicine Agency (EMA), US-Food and Drug Administration (FDA), Nation et al., Siriraj Hospital and our study regimens Table 2 and Table 3 [12,16,17,18].

### 2.4. Monte Carlo Simulation

Monte Carlo Simulation (Crystal Ball version 2017; Decisioneering Inc., Denver, CO USA) was conducted to generate 10,000 subjects for IV dosage regimens of colistin based on the linear pharmacokinetic behavior. The *f*AUC/MIC following multiple intravenous doses of various regimens was determined for the profile of each individual. Log-normal distributions were studied for between-patient variability. Probability of target attainment (PTA) was determined as the percentage of all 10,000 estimates which achieved or exceeded pharmacodynamics surrogate indices (*f*AUC/MIC ≥25) with increasing MIC [13]. The cumulative fraction of response (CFR) was calculated as the proportion of %PTA of each MIC according to the MIC distribution. The PTA and CFR of 90% were considered successful.

## 3. Results

### 3.1. Microbiology

All clinical isolates of 116 CRE include carbapenem-resistant *K. pneumoniae* (n = 96) and carbapenem-resistant *E. coli* (n = 20) were used. Colistin MIC’s ranged from 0.25- >128 mcg/mL. The overall colistin MIC_50_ and MIC_90_ were 1 and 16 mcg/mL, respectively. Among the carbapenem-resistant isolates, 91 of 116 isolates (78.44%) were susceptible to colistin (MIC **≤**2 mcg/mL). Of the colistin-susceptible isolates, 74 were *K. pneumoniae* (74/91; 81.32%) and 17 were *E. coli* (17/91; 18.68%). Colistin-susceptible *K. pneumoniae* isolates had MIC_50_ and MIC_90_ of 0.5 and 2 mcg/mL, respectively. For colistin-susceptible *E. coli* isolates, they had MIC_50_ and MIC_90_ of 0.5 and 0.5 mcg/mL, respectively. While, 25 of 116 isolates (21.56%) were resistant to colistin, of which 22 were *K. pneumoniae* (22/25; 88%) and 3 were *E. coli* (3/25; 12%). Colistin-resistant *K. pneumoniae* isolates had MIC_50_ and MIC_90_ of 16 and 32 mcg/mL, respectively. For colistin-resistant *E. coli* isolates, they had MIC_50_ and MIC_90_ of 8 and 32 mcg/mL, respectively. Minimum inhibitory concentrations (MICs) values of the study isolates are shown in Table 4.

### 3.2. Pharmacokinetic-Pharmacodynamic Simulations

#### 3.2.1. Probability of Target Attainment

The PTA analyses of various colistin regimens and creatinine clearance to achieve 90% PTA against carbapenem-resistant *K. pneumoniae* and carbapenem-resistant *E. coli* following the *f*AUC/MIC ≥25 are shown in Figure 1A–E. This figures include the %PTA for the EMA-, FDA-, Nation et al., Siriraj Hospital together with our study regimens. None of the colistin dosing regimens achieved PTA target against the MIC_90_ for colistin-resistant *K. pneumoniae* and colistin-resistant *E. coli* (32 mcg/mL) in all renal function groups. Similarly, none of the colistin dosing regimens achieved ≥90% PTA against the MIC_50_ for colistin-resistant *K. pneumoniae* (16 mcg/mL) in patients with creatinine clearance >10 mL/min. For patients with creatinine clearance ≤10 mL/min, attainment rate of 90% was observed only in our study regimen using 180 mg every 8 h. At the MIC_50_ for colistin-resistant *E. coli* (8 mcg/mL), none of the colistin approved by EMA-, FDA-, Nation et al. as well as Siriraj Hospital achieved PTA target in all renal function groups. However, our study regimens with daily dose of at least 450 mg at creatinine clearance 11-29 mL/min and daily doses of at least 360 mg at creatinine clearance ≤10 mL/min could achieve 90% PTA.

At the MIC of 0.5 mcg/mL (MIC_50_ for colistin-susceptible *K. pneumoniae* and MIC_50_/MIC_90_ for colistin-susceptible *E. coli*), all of the dosage regimens achieved ≥90% PTA regardless of renal function. At the MIC of 2 mcg/mL (MIC_90_ for colistin-susceptible *K. pneumoniae* and the current susceptibility breakpoint), none of the colistin dosing regimens achieved ≥90% PTA in patients with creatinine clearance ≥80 mL/min. However, the EMA-, FDA-, Nation et al. and Siriraj Hospital approved regimens provided PTA of 64.41, 69.2 and 79.7%, respectively. For patients with creatinine clearance 11–79 mL/min, the EMA-, FDA-, Nation et al. as well as Siriraj Hospital provided PTA of 55–89%. However, our study regimens using daily doses of 540 mg at creatinine clearance 51–79 mL/min and daily doses of at least 300 mg at creatinine clearance 11–50 mL/min could provide an attainment rate >90%. Finally, in patients with creatinine clearance ≤10 mL/min, an attainment rate of 90% PTA was observed for all dosage regimens with daily doses of at least 120 mg except the FDA-approved regimens.

The recommended dose based on the ability to achieve PTA target at various MICs with each dosing regimen are shown in Table 5 and Table 6.

#### 3.2.2. Cumulative Fraction of Response

Table 7 and Table 8 summarizes the cumulative fraction of response (CFR) for colistin against 116 carbapenem-resistant *K. pneumoniae* and carbapenem-resistant E. coli. Overall, there was lower than 90% CFR for all of regimens in patients with creatinine clearance >50 mL/min. For patients with creatinine clearance ≤50 mL/min, higher doses of at least 450 mg per day in patients with creatinine clearance 30–50 mL/min, at least 300 mg per day in patients with creatinine clearance 11–29 mL/min and at least 200 mg per day in patients with creatinine clearance ≤10 mL/min could achieve the CFR target. For colistin-resistant isolates subgroup, the currently approved regimens achieved CFR target in only 10–50%.

The CFR for colistin against colistin-susceptible *K. pneumoniae* and colistin-susceptible *E. coli* subgroup are shown in Table 7 and Table 8. For colistin-susceptible *K. pneumoniae*, patients with creatinine clearance ≥80 mL/min, attainment rate of 90% was observed only in Siriraj Hospital and our study regimens (150 mg every 8 h and 180 mg every 8 h, respectively). While, attainment rates were higher than 90% with all dosage regimens in patients with creatinine clearance 30–50 and ≤10 mL/min. For patients with creatinine clearance 51–79 and 11–29 mL/min, all dosage regimens achieved the CFR target except the FDA-approved regimens. In contrast, for colistin-susceptible *E. coli*, all dosage regimens achieved ≥90% CFR regardless of renal function.

## 4. Discussion

In our study, the MICs of colistin for CRE isolates were within the range of 0.25- >128 mcg/mL. The overall colistin MIC_50_ and MIC_90_ were 1 and 16 mcg/mL, respectively. Our study showed MIC_50_/MIC_90_ of 1/16 and 0.5/4 mcg/mL for carbapenem-resistant *K. pneumoniae* and carbapenem-resistant *E. coli*, respectively. In Thailand, a study which was performed at Siriraj Hospital [19] demonstrated MIC_50_/MIC_90_ of 32 /> 128 and 0.5/8 mcg/mL for carbapenem-resistant *K. pneumoniae* and carbapenem-resistant *E. coli*, respectively. A Global Surveillance Program study (including isolates of Enterobacteriaceae in Thailand) [20] demonstrated values of MIC_50_ and MIC_90_ at ≤0.012 and ≥4 mcg/mL, respectively. The discrepancies in MIC results among the studies could be due to the difference in prevalence of colistin resistance isolates (21.56% in our study, 71.3% in study at Siriraj Hospital [19] and <10% Global Surveillance Program study [20], respectively) as well as the percentage of *K. pneumoniae* isolates (82.75% in our study, 88.32% in study at Siriraj Hospital [19] and <50% Global Surveillance Program study [20], respectively). Our study showed the frequency of colistin resistance in *K. pneumoniae* to be higher than the frequency of colistin resistance in *E. coli* (22.9% versus 15%, respectively). In addition, the MIC_50_ and MIC_90_ values of *K. pneumoniae* isolates were higher than *E. coli* isolates. This was consistent with previous studies which described the use of high MICs for colistin in *K. pneumoniae* [19,21]

This is the first study using Monte carlo simulation to determine effective dosage regimens for colistin against carbapenem-resistant *K. pneumoniae* and carbapenem-resistant *E. coli*. When colistin was administered as monotherapy, all of colistin regimens approved by EMA-, FDA-, Nation et al., Siriraj Hospital would not be effective against MIC_90_ of colistin-resistant *K. pneumoniae* and colistin-resistant *E. coli* (32 mcg/mL), MIC_50_ of colistin-resistant *K. pneumoniae* (16 mcg/mL) and MIC_50_ of colistin-resistant *E. coli* isolates (8 mcg/mL) as shown by the PTA falling below 90%.

For isolates with MIC ≤0.5 mcg/mL (MIC_50_ for colistin-susceptible *K. pneumoniae* and MIC_50_/MIC_90_ for colistin-susceptible *E. coli*), all dosage regimens achieved ≥90% PTA regardless of renal function. This was consistent with a study by Tsala, et al. [13] which found >95% PTA to be effective against *K. pneumoniae* when the MIC were ≤0.5 mcg/mL by using colistin 300 mg CBA per day in patients with normal renal function. In contrast, there were lower success rates report with colistin against isolates with MIC of 2 mcg/mL (MIC_90_ for colistin-susceptible *K. pneumoniae* and the current susceptibility breakpoint) using all of recommendations in patients who had creatinine clearance > 10 mL/min. However, the EMA-, FDA-, Nation et al. and Siriraj Hospital approved regimens achived PTA target in approximately 65–89%. Interestingly, all the regimens with daily doses of at least 120 mg could achieve the target for patients with creatinine clearance of ≤10 mL/min. This was consistent with a study by Tsala, et al. [13] which showed lower success rates using colistin 300 mg CBA per day in patients who had normal renal function for isolates with MIC ≥2 mcg/mL.

Previous studied have shown that the EMA-, FDA- and Nation et al. approved regimens could achieve PTA target for patients with creatinine clearance <80 mL/min. Our results were in contrast to the above studies, and these results might be explained because previous studies used the average plasma concentration at steady state (Css, avg) of 2 mcg/mL which was a reasonable PK/PD target of colistin against gram negative organisms [12,16]. However, the PK/PD target of Css, avg of 2 mcg/mL was appropriate for pathogen with MIC ≤2 mcg/mL [12,16]. The prevalence of colistin resistance (MIC >2 mcg/mL) among isolates in our study was rather high (approximately 40%) and the MIC50 and MIC90 values were higher than other studies. For this reason, the PK/PD target of Css, avg of 2 mcg/mL was inappropriate to use in our study. The PK/PD targets *f*AUC/MIC≥25, which were used in this study, were determined using limited information [13]. However, several studies including in vitro studies as well as infection models which used neutropenic mice have confirmed the relation to clinical applications with these targets for colistin [13,14]. In addition, our study used higher PTA target compared with other studies (≥90% versus 80-90%).

The FDA-approved regimens have shown lower target attainment rates than other regimens. Similarly, the studies from Nation et al. [12,16] showed that FDA-approved regimens were substaintially lower than those approved by the EMA. The data has also shown that patients with renal impairment have a better chance to achieve PTA target compared with normal renal function. The findings of our study were consistent with previous analyses, the dosing suggestions of Nation et al. achieved 80–90% of patients in each of 3 groups of creatinine clearance patients (50 to <80, 30 to <50, <30 mL/min). At creatinine clearance ≥ 80 mL/min, the suggested doses only provided <40% PTA for plasma colistin Css, avg of 2 mcg/mL [12]. Additionally, renal function significantly impacts upon the proportion of colistimethate for conversion to colistin due to colistimethate being predominately renally excreted by with only small fraction of the dose converted to colistin [9,22,23].

Our results have suggested that current maintenance doses may not be effective against isolates with an MIC >0.5 mcg/mL. This is consistent with a study by Garonsik, et al. [9] and Tsala, et al. [13] which showed this MIC as the highest MIC offering reasonable PTA for current maintenance doses. Similarly, previous studies have reported that colistin monotherapy might be non-inferior when compared to combination therapy used for treating blood stream infections which are caused by carbapenem-resistant gram negative bacilli with the MIC for colistin as ≤ 0.5 mcg/mL (mortality rate 83 versus 79%, respectively) [24]. These results have suggested a susceptibility PK/PD breakpoint of ≤0.5 mcg/mL might be able to identify between isolates which are resistant and susceptible to colistin. This suggested breakpoint is less than the present breakpoints (EUCAST) value of ≤2 mcg/mL [11]. However, a prospective cohort clinical study did not find any differences in mortality compared with patients infected by colistin-susceptible and colistin-resistant CRE with 2 mcg/mL breakpoint [25]. Furthermore, clinical study by Choi, et al. [26] found that the 7-day and 14-day mortality rate in patients who were infected by colistin-susceptible *A. baumannii* (MIC < 2 mcg/mL) with high colistin MICs (≥1 mcg/mL) were statistically significant higher than ones with isolates with lower MICs (< 1 mcg/mL) (38.2% versus 20.2%, p = 0.025 and 35.2% versus 19.1%, p = 0.014, respectively). Therefore, a revision of the current colistin susceptibility breakpoint for Enterobacteriaceae may be needed.

Based on a PK/PD target of *f*AUC/MIC ≥25, CFR are lower than 90% for all regimens approved by EMA-, FDA-, Nation et al., Siriraj Hospital in all categories of creatinine clearance. Overall, they are approximately 70–86%. For colistin-resistant isolates, the currently approved regimens achieved CFR target in only 10–50%.

Our simulations were carried out on colistin-susceptible *K. pneumoniae* and colistin-susceptible *E. coli* subgroup. For colistin-susceptible *K. pneumoniae*, this study has shown an attainment rate of 90% CFR was observed in Siriraj Hospital as well as our study regimens in patients with creatinine clearance ≥80 mL/min. However, CFR targets are achieved for all dosage regimens of patients with creatinine clearance <80 mL/min except the FDA-approved regimens in patients with creatinine clearance 51–79 and 11–29 mL/min. While, all of dosage regimens achieved CFR target in all renal function groups for colistin-susceptible *E. coli* infection.

This data supports that colistin is still the mainstay treatment option against carbapenem-resistant *K. pneumoniae* and carbapenem-resistant *E. coli* infections provided that: (i) the colistin MIC for the infecting organism is ≤0.5 mcg/mL (ii) high-dose regimens are administered to attain the PK/PD targets. However, high dose of colistin may have adverse effects such as nephrotoxicity and neurotoxicity [27,28,29]. It is necessary to be careful to balance between risks of toxicity and clinical benefit (iii) for combination of the colistin and other antibiotics, the MICs were markedly lower than the MICs of the antimicrobials in monotherapy and resulting MIC values of colistin ≤0.5–1 mcg/mL.

The study has limitations as follows. First, the number of isolates evaluated was not very large and had only two species (*K. pneumoniae* (*n* = 96) and *E.coli* (*n* = 20)). The isolates came from a tertiary university hospital in Bangkok, which might be different from other hospitals or other countries. These are factors which affect the dosing regimen based on the CFR results. Second, we did not simulate about the drug level for patients using hemodialysis. Finally, the molecular characteristics of carbapenemase enzyme has not been performed in this study.

## 5. Conclusions

Our study is the first to determine effective and compare the colistin dosing regimens approved by the EMA-, FDA-, Nation, et al., Siriraj Hospital and our study against carbapenem-resistant *Klebsiella pneumoniae* and carbapenem-resistant *Escherichia coli*. It used differences between the regimens, especially for FDA- provided attainment rates which were lower than other approved regimens. The reduction in the CrCl improved attainment of %PTA and %CFR. Findings indicate the use of alternative dosing approaches allowing for higher daily doses or combination therapy are preferable for patients with CrCl ≥80 mL/min and if the MIC of colistin ≥0.5 mcg/mL.

## Figures and Tables

**Figure 1 antibiotics-08-00125-f001:**
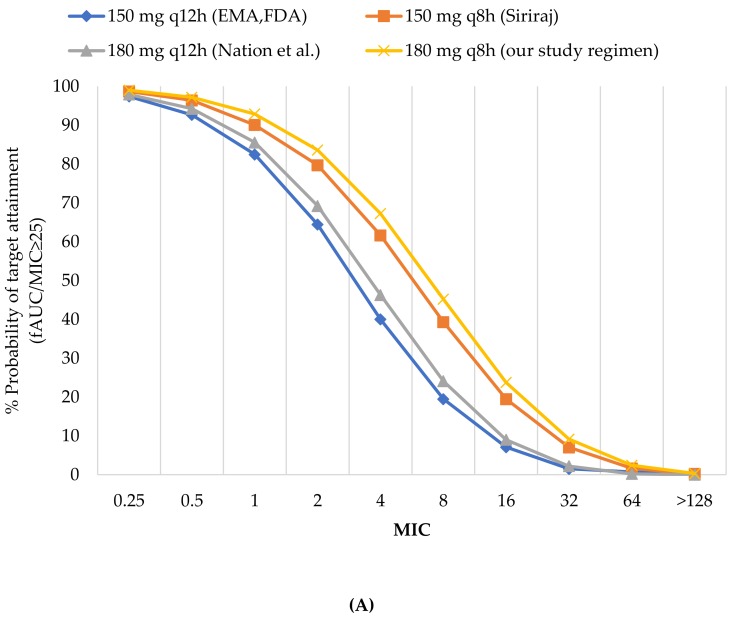
(**A**) %PTA of colistin in patients with CrCl ≥80 (mL/min). (**B**) %PTA of colistin in patients with CrCl 51–79 (mL/min). (**C**) %PTA of colistin in patients with CrCl 30–50 (mL/min). (**D**) %PTA of colistin in patients with CrCl 11–29 (mL/min). (**E**) %PTA of colistin in patients with CrCl ≤10 (mL/min). The alternative data of Figure (**D**) and (**E**) were shown in Appendix A.

**Table 1 antibiotics-08-00125-t001:** Population pharmacokinetic parameters.

Category	Parameter	Units	Estimate	%SE	%IIV
**CMS**	V1	L	12.9	-	40.4
	V2	L	16.1	-	70.9
	CLD1	L	9.57	10.5	80.1
	CLR	L/h/CrCL	0.0340	6.85	75.2
	CLNR_CMS_	L/h	2.52	3.71	39.8
**Colistin**	V3	L	57.2	5.13	43.5
	CLT_C_	L/h	3.59	-	37.9
	CLR_C_	L/h/CrCL	0.00834	27.7	-
	CLNR_C_	L/h	3.11	4.38	-

V1 = Central volume for CMS, V2 = Peripheral volume for CMS, CLD1 = Distributional clearance between the central and peripheral compartments for CMS, CLR = Renal clearance of CMS, CLNR_CMS_ = Non-renal clearance of CMS, V3 = Volume of distribution of formed colistin, CLTc = Total clearance of colistin, CLR_C_ = Renal clearance of colistin, CLNR_C_ = Non-renally of colistin clearance, %SE = The standard error or the precision of the estimates, %IIV = The inter-individual variability in the population, CrCL = creatinine clearance.

**Table 2 antibiotics-08-00125-t002:** Colistin dosage regimens for Simulations.

Creatinine Clearance	Daily Dose (CBA)	
**≥80 mL**/**min**	150 mg every 8 h	Siriraj Hospital regimen
	180 mg every 12 h	Nation, et al. regimen
	150 mg every 12 h	EMA-approved regimen FDA-approved regimen
**51 to 79 mL**/**min**	150 mg every 12 h	EMA-approved regimen Siriraj Hospital regimen Nation, et al. regimen
	114 mg every 12 h	FDA-approved regimen
**30 to 50 mL**/**min**	125 mg every 12 h	EMA-approved regimen
	110 mg every 12 h	Nation, et al. regimen
	150 mg every 24 h	FDA-approved regimen
	100 mg every 12 h	Siriraj Hospital regimen
**11 to 29 mL**/**min**	180 mg every 24 h	Nation, et al. regimen
	150 mg every 24 h	EMA-approved regimen Siriraj Hospital regimen
	60 mg every 24 h	FDA-approved regimen
**≤10 mL**/**min**	150 mg every 24 h	Nation, et al. regimen
	120 mg every 24 h	EMA-approved regimen
	60 mg every 24 h	FDA-approved regimen

**Table 3 antibiotics-08-00125-t003:** Our study regimens for Simulations.

Creatinine Clearance	Daily Dose (CBA)
**≥80 mL**/**min**	180 mg every 8 h
**51 to 79 mL/min**	180 mg every 8 h
	150 mg every 8 h
	180 mg every 12 h
	100 mg every 8 h
**30 to 50 mL**/**min**	180 mg every 8 h
	150 mg every 8 h
	180 mg every 12 h
	150 mg every 12 h
	100 mg every 8 h
**11 to 29 mL**/**min**	180 mg every 8 h
	180 mg every 12 h
	150 mg every 12 h
	150 mg every 8 h
**≤10 mL**/**min**	180 mg every 8 h
	150 mg every 8 h
	100 mg every 8 h
	180 mg every 12 h
	150 mg every 12 h
	100 mg every 12 h
	180 mg every 24 h

CBA = colistin base activity, EMA = European Medicine Agency, FDA = US-Food and Drug Administration. All of regimens were infused over 30 min. The alternative data of Table 3 were shown in Appendix A.

**Table 4 antibiotics-08-00125-t004:** Minimum inhibitory concentrations (MICs) distribution of colistin.

MIC (mcg/mL)	0.25	0.5	1	2	4	8	16	32	64	>128	MIC_50_ (mcg/mL)	MIC_90_ (mcg/mL)
**All isolates (n = 116)**	3	55	19	14	4	5	10	4	1	1	1	16
**%**	2.58	47.41	16.38	12.07	3.45	4.31	8.6	3.45	0.86	0.86		
	**Colistin-Susceptible Isolates (MIC ≤ 2 mcg**/**mL)**											
***K*** **. *pneumoniae* (n = 74)**	3	40	18	13	-	-	-	-	-	-	0.5	2
***E*** **.** ***coli* (n = 17)**	-	15	1	1	-	-	-	-	-	-	0.5	0.5
	**Colistin-Resistant Isolates (MIC > 2 mcg**/**mL)**											
***K*. *pneumoniae* (n = 22)**	-	-	-	-	3	4	10	3	1	1	16	32
***E*. *coli* (n = 3)**	-	-	-	-	1	1	-	1	-	-	8	32

The alternative data of Table 4 were shown in Appendix A.

**Table 5 antibiotics-08-00125-t005:** The recommended dose based on the ability to achieve PTA target at various MICs.

Creatinine Clearance (mL/min)	MIC 0.5 mcg/mL Daily Dose (CBA)	MIC 2 mcg/mL Daily Dose (CBA)	MIC 8 mcg/mL Daily Dose (CBA)
**≥80**	150 mg every 12 h (EMA, FDA)	Not recommended	Not recommended
**51–79**	114 mg every 12 h (FDA)	180 mg every 8 h (our study)	Not recommended
**30–50**	150 mg every 24 h (FDA)	150 mg every 12 h (our study)	Not recommended
**11–29**	60 mg every 24 h (FDA)	150 mg every 12 h (our study)	150 mg every 8 h (our study)
**≤10**	60 mg every 24 h (FDA)	120 mg every 24 h (EMA)	180 mg every 12 h (our study)

The alternative data of Table 5 were shown in Appendix A.

**Table 6 antibiotics-08-00125-t006:** The recommended dose based on the ability to achieve PTA target at various MICs.

Creatinine Clearance (mL/min)	MIC 16 mcg/mL Daily Dose (CBA)	MIC 32 mcg/mL Daily Dose (CBA)
**≥80**	Not recommended	Not recommended
**51–79**	Not recommended	Not recommended
**30–50**	Not recommended	Not recommended
**11–29**	Not recommended	Not recommended
**≤10**	180 mg every 8 h (our study)	Not recommended

MICs = minimum inhibitory concentrations (mcg/mL), EMA = European Medicine Agency, FDA = US-Food and Drug Administration, CBA = colistin base activity. The alternative data of Table 6 were shown in Appendix A.

**Table 7 antibiotics-08-00125-t007:** %PTA and %CFR for All isolates, Colistin-susceptible isolates subgroup and Colistin-resistant isolates subgroup.

	All Isolates			Colistin-Susceptible Isolates							Colistin-Resistant Isolates						
			%CFR	Colistin-Susceptible *K. pneumoniae* Subgroup			Colistin-Susceptible *E. coli* Subgroup			%CFR for all Sus-Ceptible Isolates	Colistin-Resistant *K. pneumoniae* Subgroup			Colistin-Resistant *E. coli* Subgroup			%CFR for all Resistant Isolates
	%PTA			%PTA		%CFR	%PTA		%CFR		%PTA		%CFR	%PTA		%CFR	
MIC (mcg/mL)	MIC_50_	MIC_90_		MIC_50_	MIC_90_		MIC_50_	MIC_90_			MIC_50_	MIC_90_		MIC_50_	MIC_90_		
	1	16		0.5	2		0.5	0.5			16	32		8	32		
**CrCl ≥80 mL**/**min**																	
150 mg q12 h (EMA, FDA)	82.47	7.08	70.68	92.66	64.41	85.41	92.66	92.66	90.40	86.34	7.08	1.51	12.45	19.48	1.51	20.33	13.40
180 mg q12 h (Nation et al.)	85.55	8.99	73.14	97.91	69.2	87.87	97.91	97.91	92.25	88.69	8.99	2.2	15.08	24.09	2.2	24.18	16.18
150 mg q8 h (Siriraj)	90.07	19.48	78.56	98.68	79.7	92.02	98.68	98.68	95.06	92.58	19.48	7.02	25.45	39.3	7.02	35.99	26.71
**CrCl 51 to 79 mL/min**																	
114 mg q12 h (FDA)	87.96	9.8	74.90	95.95	71.98	89.91	95.95	95.95	94.07	90.69	9.8	2.36	15.94	25.05	2.36	25.26	17.06
150 mg q12 h (Siriraj, EMA, Nation et al)	91.66	13.73	77.96	97.26	78.32	92.65	97.26	97.26	95.82	93.24	13.73	3.8	20.53	32.72	3.8	31.22	21.81
**CrCl 30 to 50 mL/min**																	
150 mg q24 h (FDA)	90.53	3.58	74.65	97.55	71.1	91..27	97.55	97.55	95.58	92.07	3.58	0.39	10.05	15.59	0.39	18.84	11.10
100 mg q12 h (Siriraj)	95.34	18.09	81.26	98.89	85.65	95.74	98.89	98.89	97.90	96.14	18.09	5.58	25	38.9	5.58	36.58	26.39
110 mg q12 h (Nation et al.)	96.29	20.03	82.17	99.03	87.43	96.35	99.03	99.03	98.19	96.70	20.03	6.41	27.05	42.66	6.41	39.01	28.48
125 mg q12 h (EMA)	97.71	23.65	83.51	99.15	89.8	97.18	99.15	99.15	98.52	97.43	23.65	8.09	30.41	46.87	8.09	42.66	31.87
**CrCl 11 to 29 mL/min**																	
60 mg q24 h (FDA)	83.41	3.57	70.63	96.33	56.2	86.26	96.33	96.33	97.91	87.56	3.57	0.42	8.11	13.11	0.42	14.41	8.87
150 mg q24 h (EMA, Siriraj)	97.92	11.16	81.45	99.67	89.35	97.44	99.67	99.67	99.61	97.73	11.16	2.02	20.20	33.24	2.02	33.27	21.77
180 mg q24 h (Nation et al.)	97.93	11.23	81.45	99.63	89.32	97.41	99.63	99.63	99.93	97.70	11.23	1.94	20.29	33.41	1.94	33.4	21.87
**CrCl****≤10 mL**/**min**																	
60 mg q24 h (FDA)	94.94	11.81	78.82	99.5	77.09	94.47	99.5	99.5	99.99	95.11	11.81	2.61	17.88	28.19	2.61	27.44	19.03
120 mg q24 h (EMA)	99.34	21.74	84.70	99.94	94.92	98.91	99.94	99.94	99.99	99.04	21.74	5.98	29.99	48.29	5.98	43.72	31.63
150 mg q24 h (Nation et al.)	99.76	28.36	86.51	99.98	97.4	99.47	99.98	99.98	100	99.54	28.36	8.69	36.24	57.76	8.69	50.55	37.96

MICs = minimum inhibitory concentrations (mcg/mL), PTA = Probability of target attainment, CFR = Cumulative fraction of response, CrCl = Creatinine clearance (ml/min), EMA = European Medicine Agency, FDA = US-Food and Drug Administration. The alternative data of Table 7 were shown in Appendix A.

**Table 8 antibiotics-08-00125-t008:** %PTA and %CFR of our study regimens for All isolates, Colistin-susceptible isolates subgroup and Colistin-resistant isolates subgroup.

	All Isolates			Colistin-Susceptible Isolates							Colistin-Resistant Isolates						
			%CFR	Colistin-Susceptible *K. pneumoniae* Subgroup			Colistin-Susceptible *E. coli* Subgroup			%CFR for all Sus-Ceptible Isolates	Colistin-Resistant *K. pneumoniae* Subgroup			Colistin-Resistant *E. coli* Subgroup			%CFR for all Resistant Isolates
	%PTA			%PTA		%CFR	%PTA		%CFR		%PTA		%CFR	%PTA		%CFR	
MIC (mcg/mL)	MIC_50_	MIC_90_		MIC_50_	MIC_90_		MIC_50_	MIC_90_			MIC_50_	MIC_90_		MIC_50_	MIC_90_		
	1	16		0.5	2		0.5	0.5			16	32		8	32		
**CrCl ≥80 mL/min**																	
180 mg q8h	92.88	23.74	80.80	97.19	83.61	93.83	97.19	97.19	96.14	92.58	23.74	9.17	29.55	45.2	9.17	40.54	30.87
**CrCl 51 to 79 mL/min**																	
100 mg q8 h	92.96	23.43	80.60	97.36	82.62	93.77	97.36	97.36	96.23	94.23	23.43	9.4	28.79	43.67	9.4	39.25	30.05
180 mg q12 h	93.43	18.04	80.15	97.82	83.54	94.31	97.82	97.82	96.72	94.76	18.04	5.7	24.88	39.23	5.7	36.31	26.25
150 mg q8 h	96.39	34.52	85.01	98.79	89.88	96.68	98.79	98.79	98.12	96.94	34.52	15.78	38.86	56.93	15.78	49.72	40.16
180 mg q8 h	97.13	39.07	86.51	99.15	92.05	97.44	99.15	99.15	98.61	97.65	39.07	19.36	43.05	61.77	19.36	54.16	44.38
**CrCl 30 to 50 mL/min**																	
100 mg q8 h	98.54	40.8	87.65	99.64	94.49	98.48	99.64	99.64	99.27	98.63	40.8	20.09	44.71	64.06	20.09	55.97	46.06
150 mg q12 h	97.9	27.61	84.91	99.41	92.44	97.84	99.41	99.41	98.91	98.04	27.61	9..97	34.47	53.71	9.97	47.36	36.01
180 mg q12 h	98.52	33.25	86.48	99.67	94.6	98.51	99.67	99.67	99.30	98.66	33.25	13.25	39.30	60.3	13.25	52.09	40.84
150 mg q8 h	99.33	54.13	90.59	99.76	97.29	99.23	99.76	99.76	99.59	99.30	54.13	29.89	55.53	76.01	29.89	65.64	56.75
180 mg q8 h	99.41	60.87	91.96	99.9	97.97	99.44	99.9	99.9	99.76	99.50	60.87	35.86	60.93	81.54	35.86	70.26	62.05
**CrCl 11 to 29 mL/min**																	
150 mg q12 h	99.76	50.76	90.39	99.91	98.35	99.60	99.91	99.91	99.81	99.64	50.76	25.25	53.34	76.03	25.25	64.55	54.68
180 mg q12 h	99.83	57.93	91.71	99.97	99.06	99.78	99.97	99.97	99.91	99.80	57.93	30.34	58.70	81.51	30.34	68.9	59.93
150 mg q8 h	99.91	75.28	94.91	99.96	99.57	99.88	99.96	99.96	99.93	99.89	75.28	52.4	72.87	91.10	52.4	80.38	73.77
180 mg q8 h	99.93	80.65	95.87	100	99.63	99.92	100	100	99.97	99.93	80.65	59.11	77.09	93.47	59.11	83.57	77.87
**CrCl≤10 mL/min**																	
180 mg q24 h	99.89	33.88	87.73	100	98.53	99.72	100	100	99.91	99.75	33.88	10.94	40.91	64.74	10.94	54.99	42.60
100 mg q12 h	99.93	59.11	92.02	99.98	99.18	99.83	99.98	99.98	99.93	99.85	59.11	34.22	59.95	81.32	34.22	70.5	61.16
100 mg q8 h	99.99	81.59	96.14	100	99.79	99.96	100	100	99.99	99.96	81.59	61.67	78.21	93.13	61.67	84.47	78.96
150 mg q12 h	99.98	71.94	94.13	100	99.84	99.98	100	100	99.99	99.97	71.94	46.32	69.86	89.74	46.32	78.00	70.84
180 mg q12 h	99.98	77.7	95.30	100	99.90	99.97	100	100	99.99	99.98	77.7	52.76	74.31	92.85	52.76	81.47	75.17
150 mg q8 h	99.99	89.51	97.76	100	99.97	99.99	100	100	100	99.99	89.51	74.47	85.46	97.26	74.47	90.46	86.06
180 mg q8 h	99.99	92.31	98.30	100	99.94	99.99	100	100	100	99.99	92.31	78.66	87.97	98.18	78.66	92.19	88.48

MICs = minimum inhibitory concentrations (mcg/mL), PTA = Probability of target attainment, CFR = Cumulative fraction of response, CrCl = Creatinine clearance (mL/min). The alternative data of Table 8 were shown in Appendix A.

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
