# Peer review of "Pharmacokinetic/Pharmacodynamic (PK/PD) Simulation for Dosage Optimization of Colistin Against Carbapenem-Resistant Klebsiella pneumoniae and Carbapenem-Resistant Escherichia coli"

_antibiotics, 2019, doi:10.3390/antibiotics8030125_

Round 1

Reviewer 1 Report

The manuscript supplies important information about colistin therapy for carbapenem-resistant K. pneumoniae and E. coli, especially colistin-resistant strains.

Minor points

1) There are two black boxes in Table 3.

2) Figures 1A through 1E, FDA graphs were indicated as diamonds with blue color and blue line, but light-green was used only in figure 1B. To help reader's understanding, I recommend that same marker form, color, and line color are used for same FDA conditions for all figures. For the same reason, it should be easily being able to distinguish between all reference data (EMA, Nations et al, Siriraj) and study regimen.

3) Lane 258: colstin should be colistin.

Author Response

Associate Professor Dr. Preecha Montakantikul

Department of Pharmacy, Faculty of Pharmacy

Mahidol University, Bangkok, Thailand 10400

66-87-100-9097 Preecha.mon@mahidol.ac.th

Antibiotics

August 16, 2019

           We wish to thank the reviewers for their insightful comments which we feel have substantially improved our manuscript. We believe we have addressed all of the comments that were raised by the reviewers and, in doing so, have crafted a paper that is more rigorous in content and clearer in presentation.

Response to Reviewers
Reviewer 1
We thank the reviewer for your thoughtful and helpful review. In the revised manuscript we have made almost all of the changes you have suggested. We copy below the reviewer’s comment/concerns (in bold), and then we describe how we have addressed each of the issues.

- There are two black boxes in Table 3.

Thank you. We deleted black boxes in Table 3.

- Figures 1A through 1E, FDA graphs were indicated as diamonds with blue color and blue line, but light-green was used only in figure 1B. To help readers understanding, I recommended the same marker form, color, and line color are used for same FDA conditions for all figures. For the same reason, it should be easily being able to distinguish between all reference data (EMA, Nations et al, siriraj) and study regimen.

Thank you. Done.

- Lane 258: colstin should be colistin.

Thank you. Done.

                          Sincerely

    Dr. Preecha Montakantikul

               Associate Professor

Department of Pharmacy, Faculty of Pharmacy

      Mahidol University, Bangkok, Thailand

Reviewer 2 Report

The results section is confusing. Authors can find a better way to write the results under different subheadings. 

Author Response

Associate Professor Dr. Preecha Montakantikul

Department of Pharmacy, Faculty of Pharmacy

Mahidol University, Bangkok, Thailand 10400

66-87-100-9097 Preecha.mon@mahidol.ac.th

Antibiotics

August 16, 2019

           We wish to thank the reviewers for their insightful comments which we feel have substantially improved our manuscript. We believe we have addressed all of the comments that were raised by the reviewers and, in doing so, have crafted a paper that is more rigorous in content and clearer in presentation.

Reviewer 2
We thank the reviewer for your thoughtful and helpful review. In the revised manuscript we have made almost all of the changes you have suggested. We copy below the reviewer’s comment/concerns (in bold), and then we describe how we have addressed each of the issues.

- The results section is confusing. Authors can find a better way to write the results under different subheadings.

Thank you. Done.

      Sincerely

    Dr. Preecha Montakantikul

               Associate Professor

Department of Pharmacy, Faculty of Pharmacy

      Mahidol University, Bangkok, Thailand

Reviewer 3 Report

1. In methodology, please explain how to carry out broth microdilution assay and how to calculate the MICs.

2. Delete the legend for Table 2A as it has been represented under Table 2B.

3. In Table 3, the first two groups of MIC should be 0.25 and 0.5 rather than 25.0 and 5.0. Additionally, the full name of MICs has been presented repeatedly above and below the table. Please delete the one at the bottom.

4. Combine two Table 4As into one and merge two 4Bs as well.

5. The “resis-tant” presented in second Table 4A and second Table 4B should be replaced by “resistant”.

6. Put the reference list in alphabetical order.

Author Response

Associate Professor Dr. Preecha Montakantikul

Department of Pharmacy, Faculty of Pharmacy

Mahidol University, Bangkok, Thailand 10400

66-87-100-9097 Preecha.mon@mahidol.ac.th

Antibiotics

August 16, 2019

           We wish to thank the reviewers for their insightful comments which we feel have substantially improved our manuscript. We believe we have addressed all of the comments that were raised by the reviewers and, in doing so, have crafted a paper that is more rigorous in content and clearer in presentation.

Reviewer 3
We thank the reviewer for your thoughtful and helpful review. In the revised manuscript we have made almost all of the changes you have suggested. We copy below the reviewer’s comment/concerns (in bold), and then we describe how we have addressed each of the issues.

- In methodology, please explain how to carry out broth microdilution assay and how to calculate the MICs.

Thank you. Done.

- Delete the legend for Table 2A as it has been represented under Table 2B.

Thank you. Done.

- In Table 3, the first two groups of MIC should be 0.25 and 0.5 rather than 25.0 and 5.0. Additionally, the full name of MICs has been presented repeatedly above and below the table. Please delete the one at the bottom.

Thank you. Done.

- Combine two Table 4As into one and merge two 4Bs as well.

Thank you. Done.

- The resis-tantpresented in second Table 4A and second Table 4B should be replaced by resistant”.

Thank you. Done.

Put the reference list in alphabetical order.

Thank you. Done.

                      Sincerely

    Dr. Preecha Montakantikul

               Associate Professor

Department of Pharmacy, Faculty of Pharmacy

      Mahidol University, Bangkok, Thailand
